# Comparison of Chromatic and Spectrophotometric Properties of White and Red Wines Produced in Galicia (Northwest Spain) by Applying PCA

**DOI:** 10.3390/molecules27207000

**Published:** 2022-10-18

**Authors:** Marina Pérez-Gil, Concepción Pérez-Lamela, Elena Falqué-López

**Affiliations:** 1Analytical Chemistry Area, Department of Analytical Chemistry and Food Science, Faculty of Sciences, Campus of Ourense, University of Vigo, As Lagoas s/n, 32004 Ourense, Spain; 2Nutrition and Bromatology Area (AA1 Group), Department of Analytical Chemistry and Food Science, Faculty of Sciences, Campus of Ourense, University of Vigo, As Lagoas s/n, 32004 Ourense, Spain

**Keywords:** Galician wines, chromatic properties, spectrophotometric properties, polyphenols, anthocyans, tannins, color

## Abstract

Wine is a complex matrix composed of numerous substances and color has an important influence on its quality and consumer acceptance. Color is affected by numerous factors such as pre-fermentation and fermentation operations, ageing, contact or addition of certain substances. In this study, different chromatic parameters were determined in 99 wines (58 red and 41 white) made from Galician (Northwest Spain) grape varieties. These parameters were obtained by using simple, rapid, and inexpensive spectrophotometric methodologies: color intensity, hue, total polyphenols content (Total Polyphenol Index TPI, Folin–Ciocalteu index, FCI), total anthocyans, total tannins, and color coordinates measured by the CIELab system. The influence of ageing in barrels (red wines) or using chips (white and red wines) on these parameters was also studied. A principal component analysis (PCA) was carried out to characterize the wines according to their chromatic characteristics. Application of PCA to the experimental data resulted in satisfactory classifications of studied white and red wines according to the cited enological practices.

## 1. Introduction

Galicia, a region located in Northwest Spain, is one of the Spanish regions with more wine “Denominations of Origin” (DO): 5 in total (“Monterrei”, “Rías Baixas”, “Ribeira Sacra”, “Ribeiro” and “Valdeorras”) (see Figure 1). However, unlike what happens with other Spanish DOs, the surface dedicated to grape crops is reduced. Approximately 10,900 ha were dedicated to vineyards in 2018 [1]. In the last decade, the wine production has been increased around 37% (mean value), although this percentage is not uniform for the 5 DOs (see Table 1).

Orography in this area has designed a viticulture based on “smallholding”, where the viticultor proportion is higher compared to the total vineyard surface. There are numerous wineries: between 27 for Monterrei DO and 179 for Rías Baixas DO. Thus, relation liter/winery moves between 45,000 (Ribeira Sacra DO) and 166,000 for Rías Baixas DO (Table 1).

All the facts showed in Table 1 carry a lower input for laboratory analyses; therefore, the availability of sophisticated equipment to analyze grapes and wines is scarce. In general, the wineries have the basic instrumentation and big wineries have usually developed conjoint research projects with universities. Normally, university research groups provide chromatographic equipment, and wineries possess spectrophotometers or colorimeter apparatus that are ease of use as a routine analytical technique [2].

Spectrophotometric methods have been extensively used in wine production to check maturity and quality parameters in grapes and wines as color, polyphenols, and their changes with different viticulture practices (grape variety, type of soil, climate, vineyard conducting systems…) and oenology treatments as yeast or enzymes addition, storage (in bottle) or aging processes (in wood barrels or with chips) [3]. These methods are adequate to be developed in small wineries that cannot afford the cost of sophisticated apparatus to test the wine quality. Moreover, these methods are simple, economic, and less time consuming than chromatographic methods (HPLC or GC), which also require previous expertise and more cost related to human resources, equipment, and other facilities.

Chromatic profile and phenolic composition of wines are increasingly used to characterize and typify them. Some authors reported that phenolics constitute a promising class of compounds used to categorize wines [4]. Color is one of the main quality parameters in a wine and variations in wine types are largely due to the concentration and composition of wine phenols [5], anthocyans being the main contributors to a red wine color [6]. In fact, for red wines, the color is very relevant for their quality [7] and for consumer acceptance [8]. Moreover, the color influences sensory properties such as flavor, taste, and aroma [9,10,11,12]. Regarding white wines, there are significantly fewer studies related to the color and the phenolic composition, in comparison with red wines [13,14]. There are a large number of typical white wines in this Spanish geographical area which have not yet been studied extensively. In particular, there are very few works related to polyphenols in white wines from the Northwest Spain region [15,16], particularly in the case of wines obtained from autochthonous white grape varieties as Godello, Albariño, Loureira, or Treixadura.

The tendency of wine to improve, or at least change during aging, is one of its more fascinating properties [17]. Normally, the aging process is used in wine to stabilize it and to improve its quality. The aging processes modify sensory properties in wine as it is accompanied by the development of color, aroma, and flavor [18]. A traditional barrel can be effectively substituted by ageing with oak chips to improve color and wood-aromas [19]. Enzyme addition is a known oenological practice that improves anthocyanins’ extraction [20] and therefore color extraction. Yeast addition is another practice that can stabilize wine color [21]. Thus, all these practices were checked to measure various chromatic and color properties in order to classify wines.

Some works have studied spectrophotometric parameters used to characterize phenolic composition and chromatic properties in Galician young red wines [22,23], but there are very few papers reporting these parameters in Galician wines subjected to aging processes [24]. Regarding other Spanish wines, very few works have reported the effect of aging on color properties and phenolic composition of white wines [25,26,27] being more numerous the ones related to red wines [28,29].

Principal Component Analysis (PCA) is a statistical tool used to find correlations between wine properties and different treatments, and has been used effectively in some works, where color properties were analysed in wines [30,31] and allowed their classification [32].

The main objectives of this work were the determination of chromatic characteristic and total polyphenols in 99 wines (58 red and 41 white) produced in Galicia by measuring spectrophotometric parameters and compare these results in order to find differences and similarities in wine profiles by PCA. Some of the studied wines were obtained by different oenological processes (aging with wood barrels or chips, addition of enzymes and yeasts), and most of them were monovarietal wines. The establishment of chromatic relations between all the parameters considered: phenolic compounds (tannins, anthocyans) spectrophotometric measurements (color intensity, CI; Tint or Hue; CIELab coordinates; total polyphenol index, TPI; Folin–Ciocalteu index, FCI) will help the winery to focus on the main measurements to typify their wines correctly and will be a quality tool in order to consider a certain variety adequate for oenological treatment or aging processes.

## 2. Results and Discussion

### 2.1. Spectrophotometric Determinations

Regarding color intensity, for red wines (Table 2), the highest values were obtained for control wine and for aged wines in oak barrels or with oak chips (samples R1–R35). The commercial samples (R36–R58) gave the lowest values, especially the ones made with Brancellao and Merenzao grapes, due to their low content of anthocyanins. Color intensity values did not differ from other red grape varieties for control samples [33,34]. Color components (yellow, red, and blue) follow a similar tendency in aged wines for yellow and blue colorations, showing low values at 3 months barrel/oak contact, increasing after 5 and 7 months contact and reaching initial values after 9 or 12 months contact (see Appendix A). In these samples, the red component is high after 3 months and decreases after 5 and 7 months to be recovered at almost initial levels after 9 or 12 months of aging. The ratio between yellow and red colorations (A420 nm/A520 nm) corresponded to the tonality or hue, which gives an estimation of the color change toward the orange tones observed in wines during aging [2]. Tint or hue values in aged wines were higher in samples aged after 5 and 7 contact months and much lower in samples aged during 9 and 12 months, similar to what happens with other grape varieties [35].

In white wines (Table 3), color intensity varies between 0.3 and 1.5—higher values in comparison with other white grape varieties [36]. One study measures color intensity as absorbance at only 420 nm, without considering the contribution of red and blue colors [37]. Logically, the yellow component was contributing to a greater degree in CI, being the highest value in our study at 65.3% (sample W36). Similarly, in other work [38], control wine samples showed the lowest color intensity values, not being significant in our results. For aged white wines, the contact with oak chips has a little influence in yellow color (values between 46.9% and 52.1%, samples W10 and W12, respectively) (see Appendix A).

Color measured by CIELab coordinates showed interesting results. Luminosity in red wines reached the highest values, oscillating between 95.5 and 97.9 in aged samples for three months, and these values were decreasing up to around 30.0 after 9 and 12 months of aging. Other authors have observed a similar effect, i.e., the wines darkened (lower L*) after aging, attributing this to their higher phenolic content [39]. Chroma data were presenting the lowest values for the red wines samples aged for 3 months.

For white wines, luminosity was quite similar in all the samples, varying from 94.5 (sample W18) to 103 (sample W33). Lower values for L coordinate were also found in other studies with white wines from other variety [40]. Regarding Chroma, in general, the samples aged with oak chips have lower values compared to commercial wines.

TPI and FCI are spectrophotometric parameters that provide winemakers with enough information about polyphenol concentration. In aged red wines, the higher values for TPI correspond to the samples with the longest contact with wood/oak-chips; this is provoked by the extraction of more polyphenols from wood. In white wines, IPT values are 10 times lower compared to red wines; contrary to red wines’ samples, IPT values are lower in aged samples with oak-chips in comparison with commercial wines.

It is well known that anthocyanins contribute to the red color of a wine and are present at a low concentration in white wines, being 50 times lower than in red wines, considering also flavonoids and catechins [41]. In our aged red wine samples, there is a global loss of anthocyans during aging, an effect also observed in other study [42], probably due to polymerization and reactions with other wine compounds.

Tannins are one of the critical classes of phenolic substances that undergo significant changes during winemaking. Total tannins in our red wines samples are higher in those samples aged for 9–12 months (10.1–14.5 g/L of cyanidin). Other study also found higher total tannins in aged wines for other grape varieties [43]. Tannins are present in low concentrations for white wines (lower than 0.15 g/L of cyanidin).

### 2.2. Principal Component Analysis

Principal component analysis is the multidimensional technique most applied in sensory profiles, as it does not require and structure on samples (wines), and the number of variables has no limit [44]. This procedure extracts the dominant patterns in the data matrix in terms of a complementary set of scores and loading plots. PCA permits us to achieve a reduction of dimensionality, a data exploration finding relationships between objects, estimating the correlation structure of the variables and investigating how many components (a linear combination of original features) are necessary to explain the greater part of variance with a minimum loss of information. When PCA is performed on autoscaled matrix data, the principal component loadings are eigenvectors of the correlation matrix [45]. Therefore, it is a proper tool to typify wines according to their chromatic properties.

PCA seeks to establish and form the parameters analyzed, if the studied wines of our region differ or resemble each other. This requires finding the parameters specific enough to enable us to characterize our wines. In other words, the aim is to establish, on the basis of the parameters analyzed, whether the wines of our region are different or similar to each other. To do this, it is necessary to find sufficiently specific parameters that allow us to characterize our wines. This differentiation is much more difficult when the aim is to differentiate among wines of the same variety and grown in bordering areas where the climatological component has a dominant role [46].

PCA explains the pattern of correlations between a set of observed variables. In this study, the 12 analyzed variables were reduced to 8 and were used for 58 red wines, and to 5 variables for 41 white ones.

#### 2.2.1. Principal Component Analysis in Red Wines

Variables used for red wines were: CI (color intensity), tint, C (chroma) and L (luminosity), TPI (Total Polyphenol Index), FCI (Folin–Ciocalteu Index), anthocyans, and tannins. The CIELab coordinates (a and b) were not considered as they are included in chroma calculation. Correlation between variables was adequate and the first discriminant functions obtained represented 80% of the total variability. Similarly to other studies for young red wines, chromatic parameters have a significant correlation with anthocyan pigments (around 0.5) [47].

A sample plot along first and second discriminant functions is showed in Figure 2. As it is observed in Figure 2, five wine groups were established, labeled as A, B, C, D, and E. Group A is formed by Mencía aged wines for 3 months in 4 tonnelleries (labels 2, 8, 12, 17 and 21) plus one control wine (label 1).

In group B, wines are placed both aged in oak barrels along 5–7 months (labels in Figure 2: 3, 9, 10, 13, 14, 18, 19, 22, 23) and aged with oak chips 3–7 months, clearly differentiated from wines aged 9–12 months, inside group C (labels 11, 15, 16, 20, 24, 25, 29, 30). In this group, there are also two commercial wines from Mencía grapes (labels 57, 58). Group D is formed mainly with commercial wines, and Group E includes two samples aged with oak-chips 7 and 9 months (labels 33, 34).

Regarding monovarietal wines, the ones obtained with Brancellao grapes are in group C (one sample is in group D, label 42) and the ones from Sousón grapes are inside group D (except one sample in C group, label 40). Wines made with Merenzao grapes are in group D (labels 36, 45). These wines share properties of aged wines. This analysis suggests that both grape variety and aging conditions clearly modify chromatic properties in the studied red wines. This analysis also permits discarding samples not sharing general properties inside each group.

Commercial samples (labels 51–58) are elaborated with more than one grape variety and most of them are placed in group D. In this case, it is difficult to establish exhaustive conclusions due to less information related to oenological treatments. It is known that parameters as color and polyphenols can be modified with different viticulture practices as grape varieties and oenological treatments or aging processes [3,48].

#### 2.2.2. Principal Component Analysis in White Wines

PCA in white wines is usually applied to characterize its aromatic profile. As far as we know, no work related to PCA involved in wine chromatic properties (measured by spectrophotometry) has yet been published. There are some articles related to polyphenolic compounds measured by chromatography [49]. They are much fewer studies related to the color and the phenolic composition in white wines in comparison with red wines. In particular, the works related to polyphenols in Galician white wines are still few, especially in the case of wines obtained from autochthonous white grape varieties as Albariño, Treixadura, Loureira, and Godello [15,16,50].

PCA in white wines showed, when calculating the correlation matrix, that TPI and L exhibited a high correlation; therefore, they were suppressed from considered variables in the analysis. Finally, only five variables were considered (FCI, CI, tint, C, and L) in 41 wine samples. The obtained plot is showed in Figure 3, where four groups were established: F, G, H, and I.

Group F contains the white wines elaborated with Godello grapes and aged with chips along 2 months, and these wines are clearly different from the others. In group G, except two samples: 19 and 41 (made with Godello and with a mixture of Godello and Treixadura grapes, respectively). Samples of group H are commercial wines from different trademarks, obtained from Godello grapes. Finally, group I agglutinate mainly wines made with Albariño grapes.

## 3. Materials and Methods

### 3.1. Wine Samples

A total of 58 red wines (coded R1–R58) and 41 white wines (coded W1–W41) from Galicia (Northwest Spain) were analyzed (Table 4 and Table 5). The wines were obtained from some native *Vitis vinifera* grape varieties collected in the five different Galician Denominations of Origin. Some samples were commercial wines obtained from supermarkets. Other samples were obtained from wineries at the industrial or semi-industrial scale from Mencía and Godello grapes, respectively, and subjected to oak-contact (with barrel or chips) and sampling at different times: 3, 5, 7, 9, and 12 months for Mencía wines (French oak barrels from 4 tonnelleries, and contact with French or American oak chips), and 7, 15, 30, and 60 days for Godello wines in contact with chips from 2 types of French oak and 1 of American oak).

### 3.2. Analytical Methods

All spectrophotometric determinations were performed, in triplicate, diluting when necessary, using a UV-vis spectrophotometer (Hitachi U-2000) with 0.1 cm or 1 cm path length glass or quartz cell, and all absorbance values were corrected to 1 cm path length.

#### 3.2.1. Color Determinations

Color intensity (CI) was determined as the sum of absorbances at 420, 520, and 620 nm [CI = A_420_ + A_520_ + A_620_] according to Glories [51]. Tint (hue or brown index) was quantified [51] as the ratio between the absorbances at 420 and 520 nm [T = (A_420_/A_520_) × 100].

CIELab coordinates were determined using a Minolta colorimeter (model CR-210). The parameters a (green-red coordinate), b (blue-yellow coordinate), and L (luminosity) were intercorrelated [52] with the chroma (C), which was calculated by the formulae: C = (a*^2^ + b*^2^)^1/2^.

#### 3.2.2. Polyphenolic Determinations

The total polyphenolic content was determined by using the Total Polyphenol Index (TPI) and Folin–Ciocalteu Index (FCI), following the methods described by Ribéreau-Gayon [53] and Zamora Marín [3], respectively.

Total polyphenol index (TPI) was measured spectrophotometrically measuring the absorbance of the wine diluted with water (100-fold for the red wines and 10-fold for the white wines) at 280 nm [TPI = A_280_ × Dilution factor].

For the Folin–Ciocalteu Index (FCI), the wine was diluted with water (5- or 10-fold) and added to the Folin–Ciocalteu reagent and Na_2_CO_3_, and then was measured by spectrophotometry at 760 nm [FCI = A_760_ × Dilution factor × 20].

#### 3.2.3. Anthocyan Determination

Total anthocyans (TA) were analyzed according to Ribéreau-Gayon and Stonestreet’s method [54]. The wine sample (1 mL) and ethanol (1 mL) were diluted with 20 mL of HCl (2%), and then divided into two tubes. Into one of the tubes, 10 mL of this mixture were mixed with 4 mL of distilled water (4 mL), and the other had 4 mL of sodium metabisulfite (at 15%, *w*/*v*) added. After 20 min of reaction, the absorbance of both tubes was measured at 520 nm (A_1_ and A_2_, respectively). The TA content is calculated as follows: TA = (A_1_ − A_2_) × 875, and expressed in mg malvidin/L.

#### 3.2.4. Tannin Determination

Total tannins (Tan) were analyzed spectrophotometrically at 550 nm in the wine diluted with water and hydrochloric acid and heated (A_550_) vs. wine diluted in the same way but not heated (A’_550_), following the method described by Zamora Marín [3]. The results are expressed as: Tan = (A_550_ − A’_550_) × 19.33.

Total tannins (as g/L cyanidin) were quantified following the Ribéreau-Gayon and Stonestreet methodology [55]. Stock solutions of cyanidin were prepared by dissolving the compound in methanol, stored at 4 °C in the darkness, and subjected to the same protocol.

### 3.3. Statistical Analysis

Statistical comparisons between both the red and white wines were made using the Student’s *t*-test, and the least significant differences (LSD) were calculated (*p* < 0.05) to determine significant differences between wines. By using the SPSS software version 19 (SPSS Inc., Chicago, IL, USA), the mean averages of all data for each type of wine (red and white) were analyzed by Principal Component Analysis (PCA), which is a multivariate technique that analyzes a data table in which observations are described by several inter-correlated quantitative dependent variables. Its goal is to extract the important information from the table, to represent it as a set of new orthogonal variables called principal components, and to display the pattern of similarity of the observations and of the variables as points in maps [56].

Discriminant analysis is the most frequently used statistical technique to classify and differentiate wines [57]. This statistical tool was usually applied to differentiate and typify wine samples, using diverse variable types as sensory data [58,59], volatile compounds [57], spectrophotometry measurements [60] and chromatographic data [61,62,63]. Therefore, it is a proper tool to typify wines according to their chromatic properties.

## 4. Conclusions

Galician wineries could only perform methodologies involving inexpensive equipment as spectrophotometers to check the grape maturity and other quality characteristic in grapes, musts, and wines. Chromatic properties in wines (red and whites) are highly influenced by varietal grape and oenological treatments like age (with chips or barrels, this last case only for red wines). In red wines, the different chromatic properties in wines made with several grape varieties are remarkable. The red color, presenting lower values after 3–5 months of wood contact, is being stabilized after 12 months, and these results follow the same tendency in total anthocyans. In general, Luminosity and Chroma are changing about one third in red wines aged in oak barrels for 12 months, in comparison to ageing for 3 months, but not in wines aged with oak-chips. Ageing with chips in white wines is not so crucial to appreciate chromatic differences. The probable different enzymes and yeast used in commercial white wines do not have a great influence on the development of color, and the PCA grouping of these samples is more disperse. Total tannins and total anthocyans are not considered adequate parameters to classify samples by PCA in white wines.

Commercial samples are also more difficult to classify both for red and white wines. In many cases, the grape used is unknown, and this fact makes the comparison and grouping difficult. Briefly, PCA using few variables (less than 8 for red wines, anthocyans, tannins, FCI, CI, tint, TPI, chroma and L–, and 5 for white wines—FCI, CI, tint, chroma and L–), obtained by simple and inexpensive methods, is an efficient statistical tool allowing for classifying/typifying wines considering ageing and discarding the samples with defects/anomalies.

## Figures and Tables

**Figure 1 molecules-27-07000-f001:**
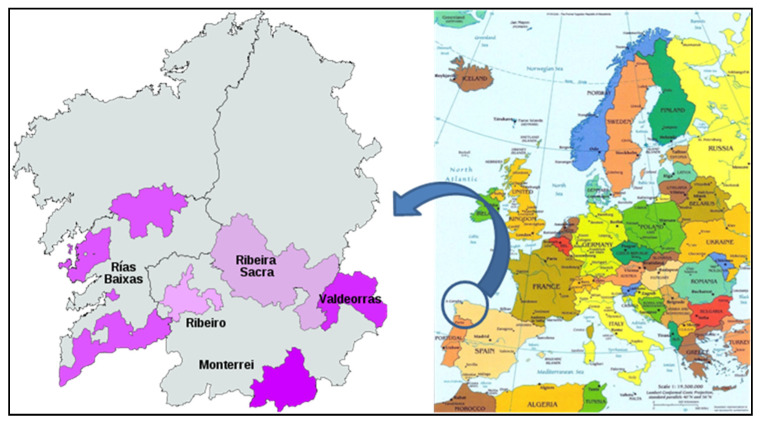
Denominations of origin from Galicia (NW Spain).

**Figure 2 molecules-27-07000-f002:**
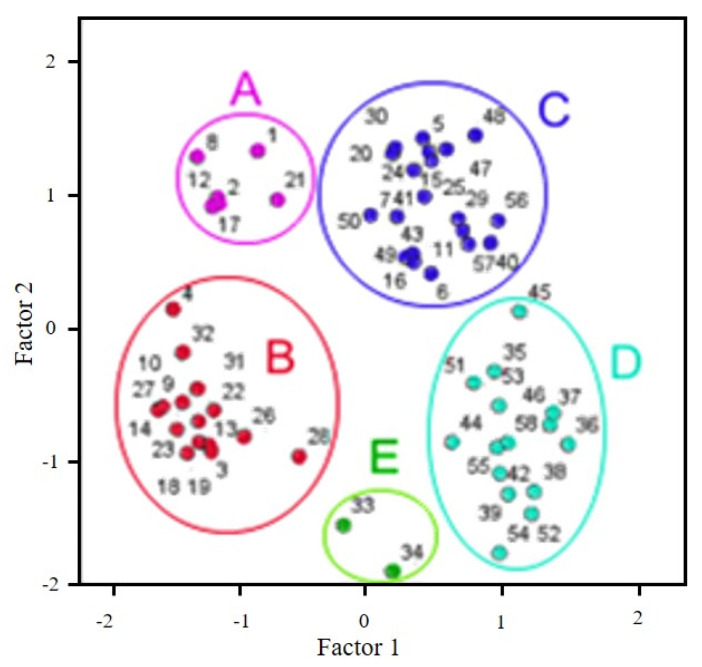
Principal Component Analysis for red wines.

**Figure 3 molecules-27-07000-f003:**
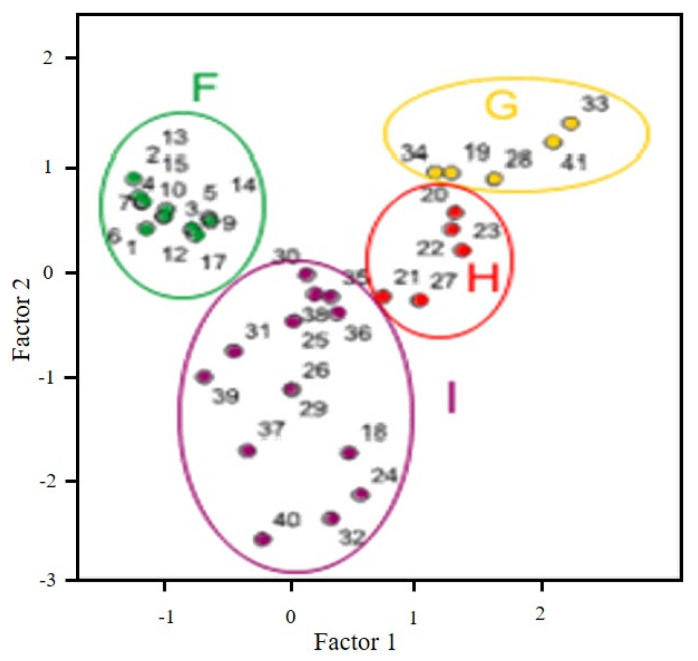
Principal Component Analysis for white wines.

**Table 1 molecules-27-07000-t001:** Galician wine denomination Origin (DO) data related to grape and wine production (year 2018).

Denomination of Origin	Monterrei	Rías Baixas	Ribeira Sacra	Ribeiro	Valdeorras	TOTAL
Surface (ha)	631	4170	2500	2500	1087	10,888
Viticultors	365	5550	3000	1667	1975	12,557
Wineries	27	179	96	103	45	450
Grape production 2021 (kg)	6,232,189	43,809,134	6,541,212	9,957,657	7,107,426	73,647,618
Liter/Winery	121,333.3	166,480.4	44,870.8	82,432.0	84,617.7	110,404.2
Liter/Viticultor	8975.3	5369.4	1435.9	5093.3	1928	3956.5
Surface/Viticultor	1.73	0.751	0.833	1.50	0.55	0.867
Winery surface	23.37	23.29	26.04	24.27	24.16	24.20
Wine production 2021 (hL)	32,760 *	298,000	43,076	84,905 *	38,078	496,819
Wine production 2010 (hL)	8466	160,665	30,758	82,816	30,848	313,553
% Increased hL	74.15	46.08	28.61	2.46	18.99	36.89

Data obtained from Spanish Ministry of Agriculture Fish and Foods (MAPA) [1], and from official web pages relative to the five DO: www.domonterrei.wine; https://doriasbaixas.com; https://ribeirasacra.org; www.ribeiro.wine; https://miconsejo.dovaldeorras.com/. (accessed on 17 October 2021) *: data from year 2020.

**Table 2 molecules-27-07000-t002:** Color and phenolic properties of red wines studied.

Wine Codes	Color		CIELab		Total Polyphenols		Anthocyans (mg Malvidin/L)	Tannins (g Cyanidin/L)
	CI	Tint	L	C	TPI	FCI		
R1	8.25 ^bcde^	63.10 ^z^	95.89 ^ab^	10.48 ^t^	59.0 ^defghijkl^	63.6 ^klmnop^	15.81 ^cd^	4.11 ^efghijklm^
R2	8.60 ^bcd^	61.69 ^z^	92.08 ^c^	3.17 ^vw^	56.6 ^hijklm^	93.0 ^defgh^	12.24 ^cd^	1.21 ^ijklm^
R3	7.04 ^cde^	460.90 ^ef^	31.61 ^mnopq^	20.84 ^nopqr^	53.4 ^klmnop^	98.7 ^cdefg^	11.37 ^cd^	1.93 ^hijklm^
R4	8.18 ^bcde^	511.70 ^b^	33.79 ^mn^	27.30 ^fghijklm^	55.2 ^ijklmno^	142.4 ^a^	4.90 ^de^	14.7 ^abc^
R5	12.34 ^a^	96.64 ^tu^	29.52 ^opqr^	16.41 ^rs^	75.0 ^ab^	60.7 ^klmnopq^	9.70 ^cd^	4.83 ^efghijkl^
R6	7.28 ^cde^	96.47 ^tu^	30.89 ^mnopqr^	22.94 ^klmnopq^	67.1 ^bcdefgh^	73.0 ^hijklm^	13.81 ^cd^	8.46 ^cdefghijkl^
R7	7.58 ^bcde^	85.40 ^wx^	29.92 ^nopqr^	23.12 ^klmnopq^	75.0 ^ab^	71.3 ^hijklm^	9.63 ^cd^	2.66 ^ghijklm^
R8	7.65 ^bcde^	63.18 ^z^	97.88 ^a^	2.36 ^w^	55.8 ^ijklmn^	125.5 ^b^	20.63 ^bc^	2.17 ^hijklm^
R9	6.38 ^cdef^	433.33 ^g^	30.41 ^mnopqr^	20.30 ^opqr^	56.8 ^ghijklm^	113.9 ^bcd^	11.63 ^cd^	11.1 ^abcdefg^
R10	7.16 ^cde^	520.95 ^a^	32.13 ^mnopqr^	29.88 ^defghij^	59.2 ^defghijkl^	72.8 ^hijklm^	10.46 ^cd^	0.72 ^klm^
R11	6.79 ^cde^	89.26 ^vw^	31.01 ^mnopqr^	24.52 ^ijklmnop^	70.2 ^abcd^	63.7 ^klmnop^	6.29 ^cde^	5.80 ^defghijkl^
R12	7.49 ^bcde^	62.50 ^z^	96.42 ^a^	8.72 ^tuv^	56.6 ^hijklm^	90.5 ^defghi^	12.63 ^cd^	3.38 ^fghijklm^
R13	6.65 ^cdef^	455.63 ^f^	33.54 ^mno^	30.56 ^cdefgh^	55.1 ^jklmno^	70.8 ^hijklm^	11.16 ^cd^	2.90 ^ghijklm^
R14	7.64 ^bcde^	491.81 ^d^	32.79 ^mno^	29.11 ^efghij^	56.7 ^ghijklm^	108.4 ^bcde^	10.55 ^cd^	2.66 ^ghijklm^
R15	10.12 ^ab^	97.51 ^tu^	31.81 ^mnopq^	24.30 ^jklmnop^	69.1 ^abcde^	70.9 ^hijklm^	9.90 ^cd^	11.8 ^abcdef^
R16	5.91 ^ef^	85.40 ^wx^	31.40 ^mnopqr^	27.39 ^fghijklm^	69.5 ^abcde^	73.5 ^hijklm^	10.11 ^cd^	0.48 ^klm^
R17	7.47 ^bcde^	61.78 ^z^	97.93 ^a^	9.08 ^tu^	54.0 ^klmnop^	104.5 ^bcdef^	11.59 ^cd^	3.14 ^ghijklm^
R18	6.34 ^cdef^	438.10 ^g^	33.54 ^mno^	31.92 ^bcdefg^	55.2 ^ijklmno^	91.0 ^defghi^	11.16 ^cd^	3.62 ^efghijklm^
R19	6.71 ^cdef^	485.03 ^d^	32.50 ^mnop^	29.61 ^defghij^	58.3 ^efghijklm^	100.2 ^cdefg^	12.46 ^cd^	3.87 ^efghijklm^
R20	8.76 ^bc^	91.41 ^uv^	30.51 ^mnopqr^	26.48 ^ghijklmn^	68.8 ^abcdef^	77.1 ^ghijkl^	8.07 ^cd^	9.42 ^cdefghij^
R21	8.07 ^bcde^	62.43 ^z^	95.48 ^ab^	9.26 ^t^	53.1 ^klmnop^	66.8 ^jklmn^	15.16 ^cd^	0.24 ^klm^
R22	6.07 ^cdef^	485.38 ^d^	32.22 ^mnopq^	27.61 ^fghijklm^	55.8 ^ijklmn^	71.8 ^hijklm^	14.72 ^cd^	2.66 ^ghijklm^
R23	6.71 ^cdef^	503.47 ^c^	32.58 ^mno^	31.27 ^cdefg^	57.6 ^fghijklm^	83.7 ^fghijk^	12.33 ^cd^	0.48 ^klm^
R24	8.27 ^bcde^	94.56 ^uv^	30.64 ^mnopqr^	24.78 ^ijklmno^	67.9 ^abcdefg^	72.1 ^hijklm^	4.46 ^de^	8.70 ^cdefghijk^
R25	10.08 ^ab^	87.35 ^w^	30.83 ^mnopqr^	24.60 ^ijklmno^	69.4 ^abcde^	65.9 ^jklmn^	9.16 ^cd^	N.D. ^m^
R26	6.41 ^cdef^	486.13 ^d^	32.54 ^mnop^	28.11 ^fghijk^	56.4 ^hijklm^	77.7 ^ghijkl^	13.16 ^cd^	7.25 ^cdefghijkl^
R27	8.07 ^bcde^	509.52 ^bc^	30.23 ^mnopqr^	18.36 ^qr^	59.5 ^defghijkl^	102.3 ^bcdef^	N.D. ^e^	3.38 ^fghijklm^
R28	8.60 ^bcd^	506.99 ^bc^	38.94 ^l^	29.41 ^defghij^	38.1 ^qr^	59.6 ^lmnopq^	0.55 ^de^	0.97 ^ijklm^
R29	8.27 ^bcde^	94.56 ^uv^	27.48 ^r^	7.41 ^tvw^	66.4 ^bcdefghi^	58.5 ^lmnopq^	10.59 ^cd^	4.35 ^efghijkl^
R30	8.44 ^bcde^	86.52 ^wx^	30.65 ^mnopqr^	21.84 ^lmnopqr^	63.9 ^bcdefghijk^	69.0 ^ijklm^	7.03 ^cde^	12.1 ^abcde^
R31	6.49 ^cdef^	511.11 ^b^	32.74 ^mno^	30.14 ^defghi^	56.9 ^ghijklm^	88.1 ^efghij^	15.37 ^cd^	3.14 ^ghijklm^
R32	7.64 ^bcde^	463.95 ^e^	32.27 ^mnop^	27.63 ^fghijkl^	54.6 ^jklmnop^	118.1 ^bc^	10.37 ^cd^	3.62 ^efghijklm^
R33	5.72 ^ef^	416.06 ^h^	45.13 ^ghij^	29.03 ^efghij^	52.7 ^klmnop^	113.2 ^bcd^	N.D. ^e^	4.59 ^efghijkl^
R34	N.A. ^h^	N.A. ^α^	39.96 ^kl^	30.64 ^cdefgh^	44.9 ^nopq^	55.8 ^lmnopq^	N.D. ^e^	N.D. ^m^
R35	4.02 ^fg^	96.86 ^tu^	41.94 ^jkl^	30.88 ^cdefgh^	44.9 ^nopq^	69.0 ^ijklm^	N.D. ^e^	3.38 ^fghijklm^
R36	0.53 ^h^	126.24 ^m^	50.49 ^f^	35.10 ^bcd^	44.8 ^nopq^	40.6 ^pqrst^	N.D. ^e^	3.87 ^e^^fghijklm^
R37	0.58 ^h^	119.72 ^no^	46.21 ^ghi^	34.04 ^bcde^	47.3 ^mnopq^	44.8 ^nopqrs^	N.D. ^e^	7.01 ^cdefghijkl^
R38	0.49 ^h^	160.13 ^j^	56.56 ^e^	35.05 ^bcd^	43.6 ^pqr^	37.8 ^qrstu^	N.D. ^e^	2.17 ^hijklm^
R39	0.51 ^h^	136.87 ^l^	48.36 ^fg^	32.65 ^bcdef^	50.8 ^lmnop^	43.9 ^nopqrs^	N.D. ^e^	6.52 ^cdefghijkl^
R40	1.04 ^h^	112.94 ^opqr^	33.33 ^mno^	23.68 ^klmnopq^	68.7 ^abcdef^	60.8 ^klmnopq^	8.29 ^cd^	5.80 ^defghijkl^
R41	1.84 ^gh^	114.51 ^nopq^	31.01 ^mnopqr^	21.29 ^nopqr^	78.5 ^a^	77.0 ^ghijkl^	8.07 ^cd^	18.8 ^ab^
R42	0.71 ^h^	132.02 ^l^	42.43 ^ijkl^	35.77 ^abc^	51.2 ^lmnop^	59.0 ^lmnopq^	N.D. ^e^	1.93 ^hijklm^
R43	1.44 ^gh^	108.78 ^pqr^	30.30 ^mnopqr^	18.75 ^pqr^	72.5 ^abc^	95.3 ^cdefgh^	9.81 ^cd^	2.17 ^hijklm^
R44	0.56 ^h^	147.96 ^k^	47.58 ^fgh^	37.53 ^ab^	55.2 ^ijklmno^	64.9 ^jklmno^	N.D. ^e^	6.04 ^defghijkl^
R45	0.80 ^h^	108.41 ^qrs^	43.13 ^hijk^	34.22 ^bcde^	36.4 ^qr^	20.6 ^tu^	0.42 ^de^	4.59 ^efghijkl^
R46	0.59 ^h^	107.47 ^rs^	43.67 ^hijk^	34.34 ^bcde^	37.8 ^qr^	22.4 ^stu^	1.81 ^de^	4.11 ^efghijklm^
R47	2.48 ^gh^	62.97 ^z^	28.21 ^qr^	10.51 ^t^	65.5 ^bcdefghij^	41.3 ^opqrst^	44.11 ^a^	14.0 ^abcd^
R48	2.40 ^gh^	65.58 ^z^	28.50 ^pqr^	11.97 ^st^	63.2 ^cdefghijk^	37.7 ^qrstu^	31.50 ^ab^	4.35 ^efghijkl^
R49	0.91 ^h^	79.86 ^xy^	34.06 ^m^	31.74 ^bcdefg^	44.1 ^opqr^	31.0 ^rstu^	7.03 ^cde^	9.67 ^cdefghi^
R50	0.92 ^h^	77.93 ^y^	33.67 ^mn^	29.45 ^defghij^	68.1 ^abcdef^	31.6 ^rstu^	7.03 ^cde^	10.4 ^bcdefgh^
R51	0.86 ^h^	101.91 ^st^	39.05 ^l^	32.92 ^bcdef^	50.4 ^lmnop^	25.1 ^stu^	N.D. ^e^	2.90 ^ghijklm^
R52	0.72 ^h^	134.22 ^l^	46.61 ^fgh^	30.99 ^cdefgh^	37.8 ^qr^	18.8 ^tu^	N.D. ^e^	5.56 ^defghijkl^
R53	0.75 ^h^	127.30 ^m^	40.22 ^kl^	31.52 ^cdefg^	51.5 ^lmnop^	23.9 ^stu^	N.D. ^e^	6.04 ^defghijkl^
R54	0.47 ^h^	170.00 ^i^	84.48 ^d^	41.06 ^a^	32.9 ^r^	16.0 ^u^	N.D. ^e^	N.D. ^m^
R55	0.70 ^h^	118.71 ^no^	38.61 ^l^	31.74 ^bcdefg^	44.1 ^opqr^	23.2 ^stu^	N.D. ^e^	1.69 ^ijklm^
R56	1.36 ^gh^	108.02 ^qrs^	32.56 ^mno^	25.30 ^hijklmno^	68.1 ^abcdef^	50.4 ^mnopqr^	2.94 ^de^	6.04 ^defghijkl^
R57	1.38 ^gh^	120.58 ^mn^	33.11 ^mno^	22.70 ^klmnopq^	69.6 ^abcd^	61.9 ^klmnop^	2.07 ^de^	10.9 ^abcdefg^
R58	0.70 ^h^	115.41 ^nop^	41.07 ^kl^	31.39 ^cdefg^	47.2 ^mnopq^	19.1 ^tu^	14.00 ^cd^	19.1 ^a^

CI: Color Intensity; L: Luminosity; C. Chroma; TPI: Total Polyphenol Index; FCI: Folin–Ciocalteu Index. N.A.: not available. N.D.: not detected. Data values in a column with different lowercase letters are statically different (*p* ≤ 0.05).

**Table 3 molecules-27-07000-t003:** Color and phenolic properties of studied white wines.

Wine Codes	Color		CIELab		Total Polyphenols		Anthocyans (mg Malvidin/L)	Tannins (g Cyanidin/L)
	CI	Tint	L	C	TPI	FCI		
W1	0.34 ^a^	182.42 ^qrst^	101.50 ^abc^	7.74 ^kl^	6.90 ^efgh^	27.7 ^a^	N.D. ^c^	N.D. ^a^
W2	0.33 ^a^	192.13 ^nop^	102.32 ^ab^	7.61 ^l^	6.46 ^gh^	19.5 ^ab^	N.D. ^c^	N.D. ^a^
W3	0.46 ^a^	169.92 ^uv^	102.15 ^ab^	8.12 ^jkl^	6.78 ^efgh^	16.2 ^b^	N.D. ^c^	N.D. ^a^
W4	0.41 ^a^	189.38 ^pq^	102.20 ^ab^	7.61 ^l^	7.01 ^efgh^	17.8 ^b^	N.D. ^c^	0.106 ^a^
W5	0.39 ^a^	185.85 ^pqrs^	102.36 ^ab^	8.00 ^jkl^	6.72 ^efgh^	14.8 ^b^	2.84 ^bc^	0.087 ^a^
W6	0.33 ^a^	185.56 ^pqrs^	102.50 ^ab^	7.48 ^l^	6.85 ^efgh^	16.9 ^b^	8.31 ^abc^	0.126 ^a^
W7	0.34 ^a^	187.23 ^pqr^	101.93 ^abc^	7.89 ^jkl^	6.64 ^efgh^	17.0 ^b^	N.D. ^c^	0.010 ^a^
W8	0.38 ^a^	176.85 ^tu^	101.72 ^abc^	8.24 ^jkl^	6.75 ^efgh^	16.9 ^b^	0.22 ^c^	N.D. ^a^
W9	0.34 ^a^	185.87 ^pqrs^	101.65 ^abc^	8.19 ^jkl^	6.22 ^h^	15.1 ^b^	5.25 ^bc^	0.145 ^a^
W10	0.34 ^a^	158.10 ^wx^	102.51 ^ab^	8.13 ^jkl^	6.54 ^fgh^	15.8 ^b^	5.91 ^sbc^	N.D. ^a^
W11	0.34 ^a^	169.31 ^v^	102.20 ^ab^	8.52 ^jkl^	6.83 ^efgh^	17.4 ^b^	N.D. ^c^	N.D. ^a^
W12	0.48 ^a^	181.16 ^rst^	101.35 ^abc^	8.77 ^jkl^	6.91 ^efgh^	15.7 ^b^	3.28 ^bc^	N.D. ^a^
W13	0.413 ^a^	171.07 ^uv^	101.96 ^abc^	8.63 ^jkl^	7.74 ^defgh^	16.7 ^b^	3.94 ^bc^	0.039 ^a^
W14	0.39 ^a^	159.83 ^wx^	102.43 ^ab^	6.56 ^l^	6.09 ^h^	16.6 ^b^	2.41 ^bc^	0.097 ^a^
W15	0.38 ^a^	157.39 ^x^	101.51 ^abc^	7.44 ^l^	6.77 ^efgh^	15.9 ^b^	8.31 ^abc^	0.155 ^a^
W16	0.45 ^a^	154.68 ^x^	102.20 ^ab^	7.53 ^l^	6.90 ^efgh^	17.7 ^b^	7.66 ^abc^	0.077 ^a^
W17	0.48 ^a^	164.79 ^vw^	101.42 ^abc^	8.03 ^jkl^	6.92 ^efgh^	16.5 ^b^	4.38 ^bc^	N.D. ^a^
W18	1.49 ^a^	223.34 ^hi^	94.54 ^e^	28.47 ^a^	11.91 ^cdefgh^	14.2 ^b^	1.31 ^bc^	0.116 ^a^
W19	0.79 ^a^	254.64 ^de^	96.16 ^de^	3.26 ^m^	8.40 ^cdefgh^	13.8 ^b^	8.75 ^abc^	0.271 ^a^
W20	0.90 ^a^	245.78 ^f^	101.24 ^abc^	17.44 ^fgh^	10.20 ^cdefgh^	13.8 ^b^	1.75 ^bc^	0.329 ^a^
W21	0.96 ^a^	265.94 ^bc^	97.51 ^cde^	18.83 ^efg^	10.04 ^cdefgh^	13.9 ^b^	N.D. ^c^	0.280 ^a^
W22	0.89 ^a^	258.80 ^c^	101.29 ^abc^	17.19 ^gh^	8.37 ^cdefgh^	14.1 ^b^	N.D. ^c^	N.D. ^a^
W23	0.71 ^a^	235.48 ^g^	101.52 ^abc^	17.18 ^gh^	10.17 ^cdefgh^	15.4 ^b^	7.00 ^abc^	N.D. ^a^
W24	1.07 ^a^	216.72 ^ij^	98.58 ^abcde^	23.99 ^c^	12.66 ^cdefg^	15.1 ^b^	N.D. ^c^	N.D. ^a^
W25	0.61 ^a^	229.87 ^gh^	101.46 ^abc^	13.72 ^i^	11.16 ^cdefgh^	16.6 ^b^	2.63 ^bc^	N.D. ^a^
W26	0.98 ^a^	208.46 ^kl^	99.82 ^abcd^	19.84 ^def^	11.96 ^cdefgh^	15.6 ^b^	N.D. ^c^	0.290 ^a^
W27	0.95 ^a^	230.95 ^g^	99.01 ^abcde^	17.47 ^fgh^	11.10 ^cdefgh^	15.4 ^b^	2.63 ^bc^	0.222 ^a^
W28	0.79 ^a^	249.25 ^ef^	99.73 ^abcd^	18.38 ^fgh^	12.23 ^cdefgh^	14.8 ^b^	9.19 ^ab^	N.D. ^a^
W29	0.96 ^a^	214.23 ^jk^	99.09 ^abcde^	21.05 ^de^	12.00 ^cdefgh^	15.8 ^b^	N.D. ^c^	N.D. ^a^
W30	0.81 ^a^	132.94 ^y^	102.58 ^ab^	10.42 ^j^	12.74 ^cdef^	14.3 ^b^	N.D. ^c^	0.232 ^a^
W31	1.31 ^a^	179.08 ^st^	100.44 ^abcd^	21.53 ^d^	13.44 ^cd^	15.8 ^b^	N.D. ^c^	0.155 ^a^
W32	1.15 ^a^	211.04 ^jk^	98.57 ^abcde^	25.45 ^b^	12.83 ^cde^	14.9 ^b^	N.D. ^c^	0.164 ^a^
W33	0.64 ^a^	257.24 ^d^	103.00 ^a^	16.22 ^h^	13.29 ^cd^	15.2 ^b^	3.06 ^bc^	0.048 ^a^
W34	0.59 ^a^	280.60 a	101.99 ^abc^	16.54 ^gh^	13.29 ^cd^	16.1 ^b^	14.44 ^a^	0.135 ^a^
W35	0.99 ^a^	191.07 ^op^	N.A. ^f^	N.A. ^n^	12.05 ^cdefgh^	14.6 ^b^	N.D. ^c^	N.D. ^a^
W36	0.93 ^a^	264.78 ^bc^	99.37 ^abcd^	21.62 ^d^	12.20 ^cdefgh^	15.7 ^b^	N.D. ^c^	0.174 ^a^
W37	1.19 ^a^	198.42 ^mn^	101.44 ^abc^	18.73 ^efgh^	25.13 ^b^	17.2 ^b^	2.19 ^bc^	N.D. ^a^
W38	0.58 ^a^	201.97 ^lm^	99.76 ^abcd^	3.85 ^m^	10.52 ^cdefgh^	14.0 ^b^	2.19 ^bc^	N.D. ^a^
W39	1.48 ^a^	214.70 ^jk^	96.01 ^de^	10.30 ^jk^	14.15 ^c^	20.8 ^ab^	5.25 ^bc^	N.D. ^a^
W40	1.00 ^a^	197.77 ^mno^	98.30 ^bcde^	2.09 ^m^	27.02 ^b^	15.6 ^b^	N.D. ^c^	0.106 ^a^
W41	0.63 ^a^	271.92 ^b^	102.57 ^ab^	17.47 ^fgh^	36.05 ^a^	14.2 ^b^	N.D. ^c^	0.087 ^a^

CI: Color Intensity; L: Luminosity; C. Chroma; TPI: Total Polyphenol Index; FCI: Folin–Ciocalteu Index. N.A.: not available. N.D.: not detected. Data values in a column with different lowercase letters are statically different (*p* ≤ 0.05).

**Table 4 molecules-27-07000-t004:** Nomenclature and characteristics of red wine simples.

Wine	Code	Treatment	Variety
Control without wood-contact (steel tank)	R1	Without wood-contact (0 month)	Mencía
	R2	Without wood-contact (after 3 months)	
	R3	Without wood-contact (after 6 months)	
	R4	Without wood-contact (after 9 months)	
	R5	Without wood-contact (after 12 months)	
	R6	Without wood-contact	
Tonnellerie-1 (French oak)	R7	Without wood-contact	Mencía
	R8	Wood-contact (during 3 months)	
	R9	Wood-contact (during 5 months)	
	R10	Wood-contact (during 7 months)	
	R11	Wood-contact (during 12 months)	
Tonnellerie-2 (French oak)	R12	Wood-contact (during 3 months)	Mencía
	R13	Wood-contact (during 5 months)	
	R14	Wood-contact (during 7 months)	
	R15	Wood-contact (during 9 months)	
	R16	Wood-contact (during 12 months)	
Tonnellerie-3 (French oak)	R17	Wood-contact (during 3 months)	Mencía
	R18	Wood-contact (during 5 months)	
	R19	Wood-contact (during 7 months)	
	R20	Wood-contact (during 12 months)	
Tonnellerie-4 (French oak)	R21	Wood-contact (during 3 months)	Mencía
	R22	Wood-contact (during 5 months)	
	R23	Wood-contact (during 7 months)	
	R24	Wood-contact (during 9 months)	
	R25	Wood-contact (during 12 months)	
Chip-1 (French oak)	R26	Wood-contact (during 3 months)	Mencía
	R27	Wood-contact (during 5 months)	
	R28	Wood-contact (during 7 months)	
	R29	Wood-contact (during 9 months)	
	R30	Wood-contact (during 12 months)	
Chip-2 (American oak)	R31	Wood-contact (during 3 months)	Mencía
	R32	Wood-contact (during 5 months)	
	R33	Wood-contact (during 7 months)	
	R34	Wood-contact (during 9 months)	
	R35	Wood-contact (during 12 months)	
Commercial wines	R36	N.S.	Merenzao
	R37–R40	N.S.	Sousón
	R41–R44	N.S.	Brancellao
	R45	N.S.	Merenzao
	R46–R48	N.S.	Sousón
	R49	N.S.	Brancellao
	R50–R51	N.S.	Mencía
	R52–R53	N.S.	Brancellao, Ferrol, Caíño longo and Caíño redondo
	R54	N.S.	Caíño tinto, Sousón and Brancellao
	R55	N.S.	Mencía
	R56	N.S.	Mencía, Merenzao and Garnacha
	R57–R58	N.S.	Mencía

N.S.: not specified.

**Table 5 molecules-27-07000-t005:** Nomenclature and characteristics of white wine samples.

Wine	Code	Treatment	Variety
Oak-chip-contact	W1	Control wine (0 days) (without chip-contact)	Godello
	W2	Control wine (7 days) (without chip-contact)	
	W3	French oak (7 days) (Type 1)	
	W4	French oak (7 days) (Type 2)	
	W5	American oak (7 days) (Type 1)	
	W6	Control wine (15 days) (without chip-contact)	
	W7	French oak (15 days) (Type 1)	
	W8	French oak (15 days) (Type 2)	
	W9	American oak (15 days) (Type 1)	
	W10	Control wine (30 days) (without chip-contact)	
	W11	French oak (30 days) (Type 1)	
	W12	French oak (30 days) (Type 2)	
	W13	American oak (30 days) (Type 1)	
	W14	Control wine (60 days) (without chip-contact)	
	W15	French oak (60 days) (Type 1)	
	W16	French oak (60 days) (Type 2)	
	W17	American oak (60 days) (Type 1)	
Commercial wines	W18–W23	N.S.	Godello
	W24–W40	N.S.	Albariño
	W41	N.S.	Godello and Treixadura

N.S.: not specified.

## Data Availability

Not applicable.

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
