# Peer review of "Comparison of Chromatic and Spectrophotometric Properties of White and Red Wines Produced in Galicia (Northwest Spain) by Applying PCA"

_molecules, 2022, doi:10.3390/molecules27207000_

Round 1
Reviewer 1 Report
The paper under review is dedicated to the comparative analysis of different chemical and spectral characteristics of an extensinve selection of Spanish wines with an emphasis on their phenolic content. The authors have done an impressive amount of work. However, I am not at all sure that this paper is a good match for such journal as Molecules, but I leave this question at the discretion of the Academic Editor of the journal. The text of the mansucript needs a significant laguage polishing.
P5 (table 5): not especified -> not specified
It would make the manuscript easier to comprehend if a more meaningful system of codes would be introduced
What is global polyphenol content? Did you mean total polyphenol content?
Section 2.2.3 and everywhere: anthocyan -> anthocyanin
Section 2.3: mean OR average, not mean average
Tables 4 and 5 do not contain the results of statistical analysis (the dispersion and significance of difference between the average values are not indicated).
Author Response
Thanks to the Reviewer 1 for his valuable comments to improve this manuscript. We provide a point-by-point response in the attached file.

Reviewer 2 Report
General comments: In this study, different chromatic parameters were determined in 99 wines (58 red and 41 white) made from Galician (Northwest Spain) grape varieties. These parameters were obtained by using simple, rapid and inexpensive spectrophotometric methodologies: color intensity, hue, total polyphenols content (Total Polyphenol Index TPI, Folin Ciocalteu index, FCI), total anthocyans, total tannins and color coordinates measured by CIELab system. The influence of ageing in barrels (red wines) or using chips (white and red wines) on these parameters was also studied. A principal component analysis (PCA) was carried out to characterize the wines according to their chromatic characteristics. The topics described in this manuscript are very interesting and have important guiding significance in the scientific community and the research of red wines. There are some light comments to the manuscript.
1. Please add line number to help review.
2. Please verify the anthocyanin determination method and calculation formula.
3. Please rewrite the statistical analysis, and add the significance analysis of the experimental results (such as Table 4 and Table 5).
4. Please modify the definition of the picture in the manuscript.
5. Results and discussion: In-depth discussion is needed and some sections need further clarification.
6. Please rewrite the conclusion.
7. The references in the manuscript are too old. Please revise them.
Author Response
We acknowledge the reviewers´ comments and their effort to make corrections and improvement suggestions. All of them have been addressed and we will answer one by one in the attached files. Moreover, we have added manuscript lines with numbers and English has been reviewed by a native person.

Round 2
Reviewer 1 Report
The authors have dealt with the issues raised in a satisfactory manner.